# WISER: SEGMENTING WATERMARKED REGION- AN EPIDEMIC CHANGE-POINT PERSPECTIVE.

## ABSTRACT

With the increasing popularity of large language models, concerns over content authenticity have led to the development of myriad watermarking schemes. These schemes can be used to detect a machine-generated text via an appropriate key, while being imperceptible to readers with no such keys. The corresponding detection mechanisms usually take the form of statistical hypothesis testing for the existence of watermarks, spurring extensive research in this direction. However, the finer-grained problem of identifying which segments of a mixed-source text are actually watermarked, is much less explored; the existing approaches either lack scalability or theoretical guarantees robust to paraphrase and post-editing. In this work, we introduce a unique perspective to such watermark segmentation problems through the lens of *epidemic change-points*. By highlighting the similarities as well as differences of these two problems, we motivate and propose WISER: a novel, computationally efficient, watermark segmentation algorithm. We theoretically validate our algorithm by deriving finite sample error-bounds, and establishing its consistency in detecting multiple watermarked segments in a single text. Complementing these theoretical results, our extensive numerical experiments show that WISER outperforms state-of-the-art baseline methods, both in terms of computational speed as well as accuracy, on various benchmark datasets embedded with diverse watermarking schemes. Our theoretical and empirical findings establish WISER as an effective tool for watermark localization in most settings. It also shows how insights from a classical statistical problem can lead to a theoretically valid and computationally efficient solution of a modern and pertinent problem.

## 1 INTRODUCTION

An unfortunate consequence of the exponential ascent of the Large Language Models (LLM), influencing all aspects of content creation, has been an increased propagation of synthetic texts across the internet. This has raised significant doubts for content authenticity and copyright infringement over multiple domains (Megías et al., 2022; Bender et al., 2021; Crothers et al., 2023; Liang et al., 2024; Milano et al., 2023; Radford et al., 2023; Chen & Shu, 2023; Woodcock, 2023), indicating an urgent need to distinguish human authorship from machine generation. "Watermarking methods" have been proposed (Christ et al., 2024; Aaronson, 2023), and widely adopted (Biden, 2023; Bartz & Hu, 2023) as a detection mechanism, embedding statistical signals into LLM-generated tokens that remain largely un-noticeable without additional information. The key insight into the watermark-based detection schemes is the use of the underlying randomness of LLM-generated outputs by incorporating pseudo-randomness into the text-generation process. When a third-party user publishes text potentially containing LLM-generated outputs with watermarks, the coupling between the LLM-generated text and the pseudo-random numbers serves as a signal that can be used for detecting the watermark. The knowledge of these pseudo-random numbers is imperative for the detection mechanism to work, making the effect of watermarking un-traceable for general users, who usually do not have access to such "keys".

Such usefulness has stimulated a plethora of research proposing myriad watermarking schemes (Kirchenbauer et al., 2024; Fernandez et al., 2023; Golowich & Moitra, 2024; Hu et al., 2024; Wu et al., 2024; Zhao et al., 2025; 2024a; Liu & Bu, 2024; Zhu et al., 2024). Concurrently, much attention has landed on the pursuit of efficient, statistically valid detection schemes (Li et al., 2025a; Kuditipudi et al., 2024; Cai et al., 2024; Huang et al., 2023; Li et al., 2024a; Cai et al., 2025). These

detection schemes usually employ the knowledge of the pseudo-random keys or deterministic hash functions to perform a composite-vs-composite test of hypotheses: $H_0$ : the entire text $\omega_{1:n}$ is unwatermarked (i.e. human generated), vs $H_1$ : the entire text $\omega_{1:n}$ is watermarked or $H_1'$ : the text $\omega_{1:n}$ contains watermarked segments. Interestingly, the literature on the more fine-grained problem of identifying/localizing the said watermarked segments, is relatively sparse; some of the available algorithms are painstakingly slow, unsuited to large texts. Moreover, to the best of our knowledge, no such algorithm to efficiently identify multiple watermarked segments has sufficient theoretical validity. This gap in the literature is also pointed out by Li et al. (2025b). In this paper, we propose WISER (**W**atermark **I**dentification via **S**egmenting **E**pidemic **R**egions): a *first-of-its-kind* computationally efficient and provably consistent algorithm to locate multiple watermarked segments from mixed-source input texts. Our method is inspired from the classical notion of *epidemic* change-points; this perspective is instrumental to both the theoretical validity and computational efficiency of our algorithm. We summarize our main contributions as follows.

### 1.1 MAIN CONTRIBUTIONS

Our key contributions are as follows.

***Novel Perspective.*** In §2, we introduce a novel, *epidemic change-point* perspective to the watermark segmentation problem by exploiting an inherent property of the watermarking schemes; see Figure 1 below. Since the segments can occur anywhere, the interpretation as an epidemic change-point enables us to re-purpose some of the classical insights into a state-of-the-art algorithm to solve a modern problem. At the same time, as discussed in §2.2.2 and 2.2.3, the particular setting of watermark segmentation introduces new challenges, and makes our analysis significantly different from the usual change-point theory.

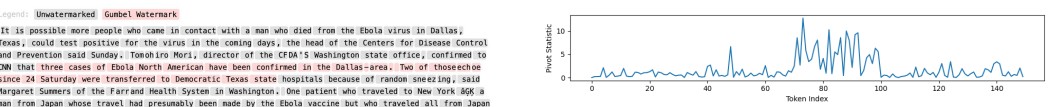

Figure 1: (Left) A mixed source text with watermarked tokens 70-100. (Right) The corresponding plot of pivot statistics vs. token.

***WISER algorithm.*** The theoretical validity of the WISER segmentation algorithm arises as an automatic consequence of our perspective. In principle, our algorithm is simple to describe; the *epidemic* interpretation produces a natural estimate for the case of only one watermarked segment; the general case of multiple watermarked segments can then be dealt with by appropriately restricting the search spaces for each of these segments. The number of such segments is estimated by a series of carefully orchestrated steps, before further restriction on the search space is ensured to lessen the computational burden. The ingenuity of our algorithm is not only in its amalgamation of different ideas from statistics, but also in its practicability. We describe the algorithm in detail in Figure 2.

***Theoretical Contribution.*** Our proposed algorithm WISER is backed by the following key result.

**Theorem 1.1** (Informal version of Theorem 3.2). *Let $\hat{I}_j, j \in [\hat{K}]$ be the output of the WISER algorithm. With explicitly mentioned choices of the tuning parameters, under standard regularity conditions, it holds that $\liminf_{n\to\infty} \mathbb{P}(\hat{K} = K, \ \max_{k\in[K]} |\hat{I}_k \Delta I_k| \approx small) \approx 1$, where, $I_j, j \in [K]$ are the true watermarked segments; $\Delta$ is the symmetric difference operator, and $K$ and $\hat{K}$ are true and estimated number of segments, respectively.*

All the theoretical results are rigorously proved in Appendix §D. Additionally, we motivate the local estimate used in the last stage of WISER by proving in Theorem 3.1 that it is consistent in the single watermarked-segment case. To the best of our knowledge, WISER is the *first watermark segmentation algorithm with complete theoretical guarantees in the most general case.*

***Computational efficiency.*** In the numerical experiments performed in §4 and Appendix §C, the theoretical guarantee shines through in WISER's superiority over the other competitive methods across different watermarking schemes and different language models. Another key aspect of its enhanced performance is its speed. WISER is specifically designed with many localized steps that

reduces its run-time, thereby making it the *only $O(n)$ watermark segmentation algorithm with provable theoretical guarantees*. We empirically also verify its speed-up in Figure 3. For a more comprehensive set of experiments and additional insights, we direct the readers to Appendix §C.

***Other contributions.*** We make some additional contributions that might be of independent interest. In terms of theory, the arbitrary dependence between the pivot statistics (introduced in §2) from the watermarked tokens, poses a significant hindrance to using the standard proof techniques from the change-point literature. Instead, we develop novel proof techniques based on moment and cumulant generating functions as well as Danskin (1967)'s results to conclude our proofs. On the application front, we address the inherent asymmetry of the watermark segmentation problem (see §2.2.3) by introducing a **M**odified **R**and **I**ndex (MRI). We argue that this provides a more accurate description of the performance of various algorithms. Due to space constraints, we have relegated both these discussions in the Appendix, in Sections D and C.1.1 respectively.

### 1.2 RELATED LITERATURE

There has been an abundance of literature on testing for existence of watermarks and the more general problems of machine-generated text detection or model equality testing (Lavergne et al., 2008; Solaiman et al., 2019; Gehrmann et al., 2019; Su et al., 2023; Mitchell et al., 2023; Huang et al., 2023; Vasilatos et al., 2023; Hans et al., 2024; Li et al., 2025a; Kuditipudi et al., 2024; Cai et al., 2024; Gao et al., 2025; Song et al., 2025; Radvand et al., 2025). However, the relatively harder problem of precisely localizing the watermarked segments from an input text has received only sparse attention. Apart from WinMax (Kirchenbauer et al., 2024), which focuses only on Red-Green watermarking, to the best of our knowledge, the only algorithms tackling the segmentation problem in its generality are Li et al. (2024b); Pan et al. (2025) and Zhao et al. (2024b). Most of these algorithms are prohibitively slow to be useful for long texts, while having little theoretical validity. In Appendix §C.1.3, we discuss the crucial limitations of each of these algorithms in contrast to WISER.

### 1.3 NOTATIONS

In this paper, we denote the set $\{1, \ldots, n\}$ by $[n]$. The $d$-dimensional Euclidean space is $\mathbb{R}^d$. We also denote in-probability convergence, and stochastic boundedness by $o_{\mathbb{P}}$ and $O_{\mathbb{P}}$, respectively. $\mathcal{L}(X)$ denotes the law of $X$. For any interval $I$, $I_L$ and $I_R$ denote its left and right end-point respectively.

## 2 WATERMARK LOCALIZATION: AN EPIDEMIC CHANGE-POINT PERSPECTIVE

Before we introduce our novel perspective in the context of locating watermarked segments, it is instrumental to establish a consistent framework of watermarking in LLM-generated texts. Let $\mathcal{W}$ denotes the dictionary, enumerated as $1, 2, \ldots, |\mathcal{W}|$. Given a text input in a tokenized form $\omega_1 \ldots \omega_{t-1}$, a watermarked LLM generates the next token $\omega_t$ in an autoregressive manner as $\omega_t = S(P_t, \zeta_t)$, where $P_t = (P_{t,w})_{w=1}^{|\mathcal{W}|}$ is the next token probability (NTP) distribution at step $t$; $S$ is a deterministic decoder function, and $\zeta_t$ is the pseudo-random variable at $t$. We grant Assumption 2.1 for the $\zeta_t$'s.

**Assumption 2.1.** *For any text $\omega_{1:n}$, there exists corresponding pseudo-random variables $\zeta_{1:n}$ available to the verifier, such that if the token $\omega_t$ at step $t$ is un-watermarked, then $\omega_t$ and $\zeta_t$ are independent conditional on $\omega_{1:(t-1)}$.*

It may seem that this assumption invalidates human edits after LLM generates a text. However, in Appendix §A, we discuss how Assumption 2.1 applies even to the mixed-source texts.

### 2.1 PIVOT STATISTICS AND ELEVATED ALTERNATIVES

Note that, a text $\omega_{1:n}$ with $K$ disjoint watermarked intervals $I_1, \ldots, I_K, I_j \subset [n]$ for $j \in [K]$, can be modeled as

$$w_t \sim \begin{cases} P_t, t \notin I_0 := \cup_{l=1}^{K} I_k \\ S(P_t, \zeta_t), \text{ otherwise,} \end{cases} \quad t = 1, 2, \ldots, n. \tag{2.1}$$

We are interested in the statistical problem of estimating the individual intervals $I_1, \ldots, I_K$ as well as $K$. Before proceeding further, it is appropriate to formally introduce the notion of pivot statistics.

**Definition 2.1.** *$Y(\omega, \zeta)$ is called a pivot statistic if $\mathcal{L}(Y)$ is same for all $\omega \in \mathcal{W}$.*

Pivot statistic has been extremely effective in providing statistically valid testing strategies for the existence of watermarks in mixed-source texts (Li et al., 2025a; 2024a; Cai et al., 2024), however, in what follows, we will demonstrate their effectiveness in aiding a localization algorithm. This effectiveness is a result of a simple property of the pivot statistics; they metamorphose the conditional independence of $\omega_t$ and $\zeta_t$ for un-watermarked tokens into $P_t$-independent distributions. Formally, this property is described in the following result.

**Lemma 2.2.** *If $S$ denotes the set of un-watermarked tokens, then $\{Y_t\}_{t \in S}$ are i.i.d.*

This ancillarity is heavily used in all the available statistical analysis of watermarked schemes; nevertheless, for the sake of completion we provide a proof in Appendix §D.3. Lemma 2.2 enables us to use the notation $\mu_0 := \mathbb{E}_0[Y(\omega, \zeta)]$ as the expectation of the pivot statistic $Y$ when the token $\omega \sim P$ is not watermarked; on the other hand, $\mathbb{E}_{1,P}[Y(\omega, \zeta)]$ will denote expectation with respect to the randomness of $\zeta$ (i.e. conditional on $P$) when $\omega$ is watermarked according to $(S, \zeta)$-mechanism. Finally, we denote $Y_t := Y_t(\omega_t, \zeta_t)$. Note that since $Y_t$ is a pivot statistic, so is $h(Y_t)$ for any *score function* $h : \mathbb{R} \to \mathbb{R}$. Usual tests for watermark detection look at $\sum_{t=1}^{n} h(Y_t)$ as a statistic for a one-sided test, and put considerable effort into constructing an effective score function $h$ (Kirchenbauer et al., 2024; Zhao et al., 2024b; Li et al., 2025a; Cai et al., 2025). Intrinsic to this construction, even though never explicitly stated, is the assumption that $\mathbb{E}_{1,P}[h(Y)]$ is usually larger than $\mu_0$ for any possible NTP distribution $P$. This hypothesis of "elevated alternatives" can also be empirically viewed in Figure 1.

We formalize this observation with the following hypothesis.

**Assumption 2.2** (Elevated Alternatives Hypothesis). *Assume that the next token distribution (NTP) $P$ belongs to a distribution class $\mathcal{P}$. Then, there exists $d > 0$ such that $\inf_{P \in \mathcal{P}} \mathbb{E}_{1,P}[h(Y)] \geq \mu_0 + d$, where $\mathbb{E}_{1,P}(\cdot) = \mathbb{E}_1[\cdot | P]$ denotes the unknown distribution of $h(Y)$ when watermarking is implemented on the NTP $P \in \mathcal{P}$.*

This assumption entails that the pivot statistics is effective conditional on any possible NTP from the class $\mathcal{P}$, ruling out trivial cases such as $Y(\omega, \zeta) \equiv \zeta$. Most standard watermarking schemes satisfy Assumption 2.2; see §B for some concrete examples. To summarize, the pivot statistics $Y_t$ has a mean level $\mu_0$ when the token $\omega_t$ is un-watermarked; on the other hand, we expect the pivot statistics to take comparatively larger values inside the watermarked segments. Interestingly, this observation establishes a ready-made connection to the notion of "epidemic change-points", sporadically explored in the classical time-series literature for the past few decades. We discuss this novel perspective in the following section.

## 2.2 WATERMARK AND EPIDEMIC CHANGE-POINT

We first provide some background on epidemic change-points, given their relative obscurity, for the convenience of readers who may be unfamiliar with the concept.

### 2.2.1 WHAT IS AN EPIDEMIC CHANGE-POINT?

An epidemic change-point refers to a situation where a stochastic process deviates in one of its features in an interval and returns to the baseline. The simplest and yet the most popular formulation of a 'mean-shift' epidemic model is as follows. Consider the time-series $X_i = \mu_i + Z_i$, where $Z_i$ is mean-zero stationary process and

$$\mu_i = \mu \text{ if } i \in \{1, \cdots, p\} \cup \{q+1, \cdots, n\} \text{ and } \mu_i = \mu + \delta \text{ if } i \in \{p+1, \cdots, q\} \qquad (2.2)$$

The epidemic change-point framework originated with Levin & Kline (1985), who studied the testing for existence of such epidemic patches for epidemiology applications, with a more comprehensive discussion in Yao (1993); Inclán & Tiao (1994). Later on, Hušková (1995); Csörgö & Horváth (1997); Chen et al. (2016) have discussed consistency, asymptotic theory as well as statistical powers of these epidemic estimators and accompanying tests. Other related papers discussing inference tailored to epidemic alternatives can be found in Račkauskas & Suquet (2004; 2006); Ning et al. (2012). Compared to the vast literature for usual change-point analysis, the epidemic change-point literature has been quite sparse, and even then, the focus has remained mostly on testing for the existence of such temporary departure rather than on locating these patches with provable statistical guarantees.

#### 2.2.2 EPIDEMIC CHANGE-POINT WITH IRREGULAR SIGNALS

Note that, in the watermarked patches, it is unrealistic to assume a fixed mean of the pivot statistics, since the next token probability distribution usually changes at each step. Therefore, results pertaining to model (2.2) are not directly applicable here. However, invoking Assumption 2.2, we can assume that the means of the pivot statistics are separated from the null by at least some margin. This puts us in a position to solve an epidemic mean-shift problem of a new kind. Very recently Kley et al. (2024) proposed usual change-point detection under the presence of such irregular signals. Concretely, for noisy data of the form $X_t = \mu_t + Z_t$, $t = 1, \ldots, n$ where $\mu_t$ are means or signals and $(Z_t)_{t \in \mathbb{Z}}$ is a stationary mean-zero noise, they considered the following hypothesis testing problem with irregular 'non-constant-mean' alternative:

$$H_0 : \mu_1 = \cdots = \mu_n \text{ vs. } H_1 : \exists \tau \in \{2, \ldots, n\}, d > 0 : \mu_1 = \cdots = \mu_{\tau-1}, \quad \mu_\tau, \ldots, \mu_n \geq \mu_1 + d.$$

They also proposed an estimation procedure for the location parameter $\tau$. In this work, in the light of the mean pattern of the pivot statistic corresponding to the watermarked region, we extend their estimators to the epidemic alternative. Moreover, the intrinsic dependence introduced by the context of how an LLM token sequence is generated also makes our premise for the error specification quite novel and thus brings out significant technical challenges.

#### 2.2.3 A SUBTLE DIFFERENCE WITH CHANGE-POINT PROBLEM

Although watermark segmentation closely resembles epidemic change-point detection, a crucial difference arises in algorithm evaluation. Standard change-point problems are symmetric; under model (2.2), the edge cases $p = 1, q = n$ and $p = q$ are equivalent. On the other hand, watermarking problems exhibit asymmetry; the edge cases (i) "the entire sequence is un-watermarked" and (ii) "the entire sequence is watermarked", differ due to irregular means of the pivot statistics under watermarking. In fact, the widely popular Rand Index (RI) - being borrowed from clustering literature, and used in watermark segmentation (Li et al., 2024b; Pan et al., 2025) - fails to capture this distinction. For the interested readers, we address this by introducing a **M**odified **R**and **I**ndex (MRI), and demonstrate its advantages over RI in Appendix §C.1.1.

### 3 THEORY FOR WATERMARK LOCALIZATION

In this section, we develop our algorithm by proceeding step-by-step. In the §3.1, we propose an estimator to localize a single watermarked segment inside a text, and establish its theoretical consistency with finite sample results. Building on this estimate, in §3.2 we formally propose the WISER algorithm. Subsequently, we theoretically establish its consistency in segmenting multiple watermarked patches, while also discussing its linear-time computational complexity.

#### 3.1 SEGMENTING SINGLE WATERMARKED PATCH

Let us denote $X_t = h(Y_t)$. Recall Lemma 2.2, the notation $\mu_0 = \mathbb{E}_0 X_t$, and Assumption 2.2. Let $\tilde{d}$ be such that there exists $\rho \in (0, 1)$ satisfying $d > 2\rho\tilde{d}$. Based on our discussion in §2.2.2, we adapt the estimator from Kley et al. (2024) for our particular setting.

$$\hat{I} = \underset{s,t \in [n]}{\arg\min} \sum_{k \notin [s,t]} (X_k - \mu_0 - \rho\tilde{d}). \tag{3.1}$$

The following theorem analyzes its convergence properties for the case of a single, un-interrupted watermarked region. Subsequently, we discuss some of its connotations in successive remarks.

**Theorem 3.1.** *Let $\{X_t\}_{t=1}^n := \{h(Y_t)\}_{t=1}^n$ be the pivot statistics based on the given input text, and assume that $I_0 \subset \{1, \ldots, n\}$ is the only watermarked interval. Grant Assumption 2.2. Denote*

$$\varepsilon_t = \begin{cases} X_t - \mu_0, t \notin I_0, \\ X_t - \mu_t, \ \mu_t := \mathbb{E}_{1,P_t}[X_t], t \in I_0. \end{cases}$$

*Suppose the class of distributions $\mathcal{P}$ is closed and compact, and there exists $\eta > 0$ such that $\sup_{P \in \mathcal{P}} \mathbb{E}_{1,P}[\exp(\eta|\varepsilon|)] < \infty$. Moreover, assume that $\min\{\mathrm{Var}_0(\varepsilon), \sup_P \mathrm{Var}_{1,P}(\varepsilon)\} > 0$. If there exists a constant $c > 0$ such that $\tilde{d} \geq c$, then $|\hat{I} \Delta I_0| = O_{\mathbb{P}}(\tilde{d}^{-1})$. Here $\Delta$ is the symmetric difference operator and $O_{\mathbb{P}}$ hides constants independent of $n, \tilde{d}, \rho$, and $\mu_0$.*

The $O(\tilde{d}^{-1})$ rate can further be sharpened to $O(\tilde{d}^{-2})$ under a local sub-Gaussianity condition (see Proposition 1 in the Appendix §D ). In fact, under very mild conditions, Theorem 3.1 already tackles a more general scenario compared to the only other theoretical result available in a similar context (Li et al., 2024b). In contrast to a general watermarked patch, Li et al. (2024b) considered a specialized scenario, where only the first half of the text till an arbitrary point is watermarked, reducing the problem to a classical change-point setting.

The parameter $\tilde{d}$ serves as the *signal strength* in the convergence diagnostics of $\hat{I}$. It allows $\hat{I}$ to look for intervals such that the $\tilde{d}$-biased mean outside that interval is minimized. To ensure accuracy, $\tilde{d}$ has to be large, but $\tilde{d} \gg d$ might lead to overestimation. Since the minimum separation $d$ in Assumption 3.1 is typically unknown, it cannot be used directly. In most cases (see examples in Appendix §B), a distribution-dependent lower bound $d_{\mathrm{L}} \leq d$ may be available, but relying on $\tilde{d} = d_{\mathrm{L}}$ often sacrifices power, as $\inf_{t \in [n]} \mathbb{E}_{1, P_t}[X_t - \mu_0]$ is usually much larger. Thus, a key step in practice is a data-driven yet valid choice of $\tilde{d}$, which we discuss in §3.2. The tuning parameter $\rho$ adjusts the impact of $\tilde{d}$ and mitigates small errors in its selection. Choosing $\rho \approx 0$ is undesirable, as it causes $\hat{I}$ to overestimate $I$ due to fluctuations above $\mu_0$ under the null. Conversely, setting $\rho \approx 1$ can violate the requirement $d > 2\rho\tilde{d}$ when $\tilde{d}$ is large. Empirically, $\rho \in [0.1, 0.5]$ provides robust performance, and we revisit these choices in our discussion of WISER as well as the ablation studies in Appendix §C.3.

*Remark* 3.1 (Connection with other performance metric). Even though Theorem 3.1 controls the estimation error in terms of symmetric difference between estimated and true watermarked patches $\hat{I}$ and $I$ respectively, it is straightforward to transform this result in terms of the more familiar Intersection-Over-Union metric $\mathrm{IOU}(I, \hat{I}) = |I \cap \hat{I}|/|I \cup \hat{I}|$ as $1 - \mathrm{IOU}(I, \hat{I}) = \frac{|I \Delta \hat{I}|}{|I \cup \hat{I}|} = O_{\mathbb{P}}\left(\frac{1}{|I|\tilde{d}}\right)$. As the text size increases ($n \to \infty$), if $|I| = O(1)$, then the number of un-watermarked tokens is too large, overpowering the signal from the watermarked tokens. Under this "heavy-edit" regime, no non-trivial test statistic can differentiate between $H_0$ : the entire text $\omega_{1:n}$ is un-watermarked (i.e. human-generated) and $H_1$ : the entire text $\omega_{1:n}$ is watermarked, with reasonable power (Li et al., 2025b). The estimation being a harder problem than testing, it is therefore reasonable to assume $|I| \to \infty$ as $n \to \infty$. Therefore, Theorem 3.1 essentially entails that $\mathrm{IOU}(I, \hat{I}) \to 1$ as $n \to \infty$.

Despite the attractive theoretical properties of $\hat{I}$ given in (3.1), notwithstanding the yet unclear choice of $\tilde{d}$, there are a couple of practical roadblocks to deploying $\hat{I}$. Firstly, $\hat{I}$ has a computational complexity of $O(n^2)$, which is quite prohibitive for a large body of text one usually encounters. Secondly, it is not straightforward as to how $\hat{I}$ can be generalized to localize multiple watermarked segments. We answer these questions by proposing our WISER algorithm.

### 3.2 WISER: SEGMENTING MULTIPLE WATERMARKED PATCHES

The main motivation behind our proposed algorithm WISER is to use the estimator $\hat{I}$ on localized disjoint intervals that are more-or-less guaranteed to contain the true watermarked segments. Such intervals with guarantees are usually recovered as a consequence of some first-stage screening. For the convenience of readers, the detailed algorithm, along with a schematic diagram of WISER containing the key steps, is illustrated in Figure 2.

Subsequently, we make a mild assumption that the true watermarked segments have a minimum length, and are also well-separated to be considered as distinct segments. Formally, for two disjoint intervals $I_1 = (I_{1,L}, I_{1,R})$ and $I_2 = (I_{2,L}, I_{2,R})$, let $d(I_1, I_2) := \min\{|I_{1,L} - I_{2,R}|, |I_{1,R} - I_{2,L}|\}$.

**Assumption 3.1** (Minimum separation). *Let $K$ be the number of true watermarked segments, with the segments themselves denoted by $I_j, j \in [K]$. Then there exists a constant $C_0 > 0$, such that $\min_{k \in [K]}\{|I_k| \wedge d(I_k, I_{k-1})\} \geq C_0 n^{1/2+\gamma'} \log n$ for some $\gamma' > 0$.*

In what follows, we explain the step-by-step rationale behind the algorithm. For clarity, we ignore the niceties of $\lfloor \cdot \rfloor$'s and $\lceil \cdot \rceil$'s.

- **Blocking stage.** Let $b = \sqrt{n}$ and the threshold $\mathcal{Q}$ be given. In the first stage, we partition the data into $\sqrt{n}$ consecutive blocks, each of size $\sqrt{n}$. Among these, we retain only those blocks for which the corresponding sum of pivot statistics exceeds $\mathcal{Q}$. Typically, to avoid multiple testing issues, $\mathcal{Q}$ is chosen as the $(1 - \alpha)$-quantile of the *null* (i.e. when there is no watermarking in the entire text) distribution of the maximum block sum over all $\sqrt{n}$ blocks.

- **Discarding stage.** Under Assumption 3.1, by definition of $\mathcal{Q}$, $O(\sqrt{\log n})$ successive unwatermarked blocks will have sum exceeding $\mathcal{Q}$ *only* with vanishing probability. Therefore, any connected interval of selected blocks from the first stage, with length at most $c\sqrt{n \log n}$, must necessarily be spurious. Hence, at this stage, we join consecutive selected blocks, and discard any connected intervals smaller than $c\sqrt{n \log n}$.

- **Enlargement stage.** The above two steps ensure $\hat{K} = K$ with probability approaching 1. Also, the intervals from the previous stage are almost accurate estimates of the true segments, except for some additional watermarked regions that were part of discarded blocks. Because of Assumption 3.1 and the size of the discarded blocks, such regions have size at most $O(\sqrt{n})$. Therefore, we enlarge each interval by $\asymp n^{1/2}$ for a small $0 < \gamma \ll 1/2$. These enlarged intervals $D_j$'s remain disjoint with high probability due to Assumption 3.1, and are therefore each amenable to (3.1) to yield $\hat{I}_j$'s.

- **Estimating $\tilde{d}$.** To estimate $\tilde{d}$, we take the sample mean of $(X_t - \mu_0)$ over $\cup_{j=1}^{\hat{K}} D_j$. This serves as a proxy for the oracle average of $(X_t - \mu_0)$ over $\cup_{j \in [K]} I_j$, which may overestimate $d$. We choose $\rho$ to calibrate it so that $d > 2\rho\tilde{d}$.

- **Reducing computational cost.** We alleviate the increased computational aspect of a naive implementation of (3.1) by leveraging additional information from the screening stage to reduce the search space. Indeed, due to our blocking and discarding steps, it can be guaranteed with high probability that, for each $j \in [K]$, $D_{j,L}$ is at most $\asymp \sqrt{n}$ distance apart from $I_{j,L}$; similarly $D_{j,R}$ is also at most $\asymp \sqrt{n}$ distance apart from $I_{j,R}$. Therefore, from $D_j$ we can produce search intervals $L_j, R_j$ of lengths $\asymp n^{1/2}$ such that $I_{j,L} \in L_j$ ad $I_{j,R} \in R_j$ with high probability, and restrict the search to $s \in L_j, t \in R_j$. Consequently, now each implementation of this modified (3.1) (see Figure 2) takes $O((n^{1/2})^2) = O(n)$ amount of computational time, leading to a speed-up while maintaining theoretical validity.

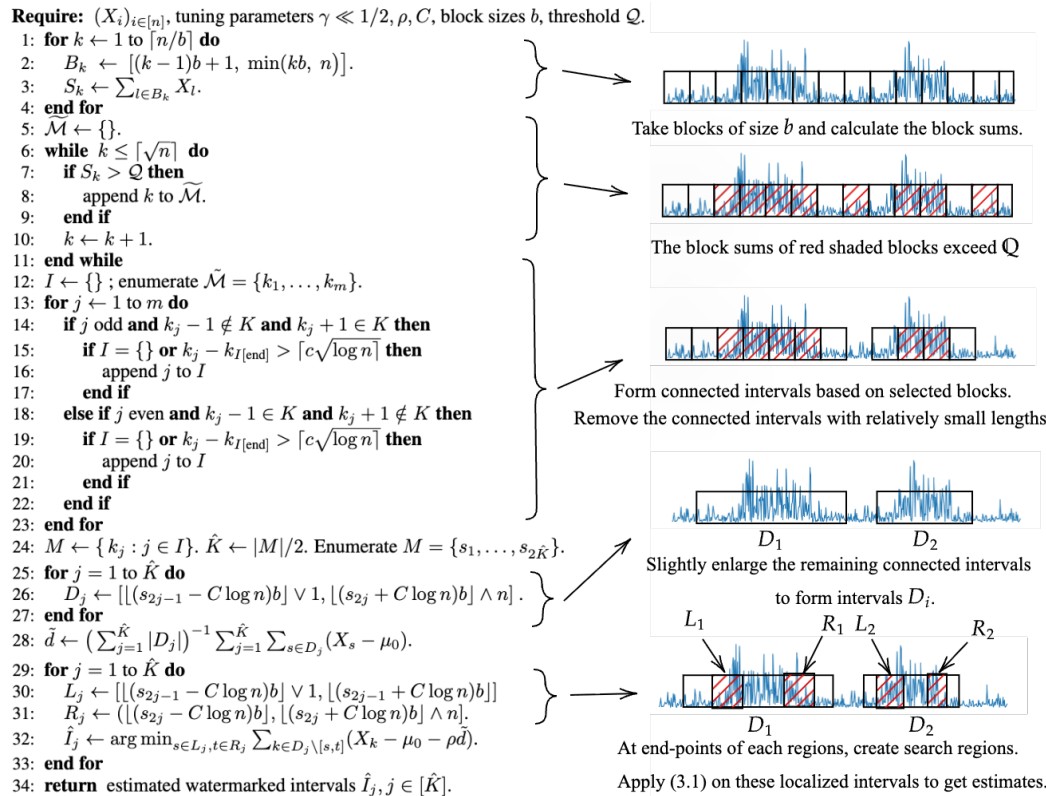

Figure 2: (Left): The Algorithm `WISER`; (Right) `WISER` in action with key steps.

The following result summarizes these insights into a formal consistency guarantee.

**Theorem 3.2.** *Assume that the null distribution of the pivot statistics is absolutely continuous with respect to the Lebesgue measure. Fix $\alpha \in (0,1)$, and recall the quantities defined in* `WISER` *described*

*in Figure 2. Let the block length $b = b_n$ satisfy $b_n \asymp \sqrt{n}$, and suppose the threshold $\mathcal{Q} = \mathcal{Q}_n$ is selected so that $\mathbb{P}_0(\max_{1 \le k \le \lceil n/b \rceil} S_k > \mathcal{Q}) = \alpha$. Also assume that $\mathbb{E}_0[|X - \mu_0|^{3+\delta}] < \infty$ for some $\delta > 0$. Let $\sup_{P \in \mathcal{P}} \mathbb{E}_{1,P}[X] < \infty$, and assume there exists $\tau > 0$ such that*

$$\kappa := \inf_{\theta \ge 0} \theta(\mu_0 + \tau d) + \log \sup_P \mathbb{E}_{1,p}[\exp(-\theta X)] < 0. \tag{3.2}$$

*Additionally, let the number of watermarked intervals $K$ be bounded, and Assumption 3.1 be granted for the watermarked intervals $I_k, k \in [K]$. Then, given $\varepsilon > 0$ and $d \ge c$ for some constant $c > 0$, under the assumptions of Theorem 3.1, there exists $M_\varepsilon \in \mathbb{R}_+$, independent of $n, K,$ and $d$, such that,*

$$\liminf_{n \to \infty} \mathbb{P}\big(\hat{K} = K, \ \max_{k \in [K]} |\hat{I}_k \Delta I_k| < M_\varepsilon d^{-1}\big) \ge 1 - \varepsilon. \tag{3.3}$$

*Remark* 3.2. We briefly discuss arguably the only technical condition (3.2) in Theorem 3.2. This can be construed as a Donsker-Varadhan strengthened version of Assumption 2.2. For an appropriate choice of the score function $h$ and some NTP distribution $P^\star$ depending on $\mathcal{P}$, the Donsker Varadhan representation (Donsker & Varadhan, 1983) entails

$$\inf_{\theta \ge 0} \theta \mu_0 + \log \sup_{P \in \mathcal{P}} \mathbb{E}_{1,p}[\exp(-\theta X)] = -D_{\mathrm{KL}}(\mathcal{L}_0(X), \mathcal{L}_{1,P^\star}(X)),$$

where $D_{\mathrm{KL}}$ denotes the Kullback-Leibler divergence, $\mathcal{L}_0$ denotes the law of un-watermarked pivot statistics, and $\mathcal{L}_{1,P^\star}$ denotes the law of watermarked pivot statistics when the NTP is $P^\star$. In light of this, $\kappa$ lifts the minimum separation between the un-watermarked and watermarked distributions into a gap between the cumulant functions, and can therefore be understood to be mild. Equation (3.2) establishes a weak uniform control over the behavior of pivot statistics under watermarked segments. This allows us to rigorously bypass the possibly arbitrary and strong dependence across the pivot statistics corresponding to watermarked tokens while deriving Theorem 3.2.

We reiterate that with $b \asymp \sqrt{n}$, WISER has a run-time only of $O(n)$ ignoring log factors. This, to the best of our knowledge, is among the *least computationally expensive* algorithms available in the literature. In view of its theoretical validity under very general conditions, this makes it a useful tool for practical applications. We showcase it through a series of extensive numerical experiments.

# 4 NUMERICAL EXPERIMENTS

Building on the theoretical validation established in the previous sections, in this section we undertake an empirical evaluation of the proposed WISER method, demonstrating its superiority over existing state-of-the-art (SOTA) algorithms. In §4.1, we compare its accuracy against competitive methods on a benchmark dataset across multiple watermarking schemes, and in §4.2, we assess its computational efficiency. Due to space constraints, we provide additional numerical experiments in Appendix §C. We encourage the readers to check it out for more practical insights, including, **(i)** a detailed explanation of the benefits of WISER over other SOTA algorithms (§C.1.3), **(ii)** experiments quantifying the effect of watermark intensity and length across different algorithms (§C.2), and **(iii)** an ablation study (§C.3) highlighting the stability of our method across tuning parameter choices.

## 4.1 COMPARATIVE PERFORMANCE OF WISER

Within the relatively limited body of literature on the identification of watermarked segments from mixed-source texts, Aligator (Zhao et al., 2024b), SeedBS-NOT (Li et al., 2024b) and Waterseeker (Pan et al., 2025) algorithms have emerged as the leading methods, producing the most accurate results so far. For an extensive comparison, our experimental setup involves completion of randomly selected 200 prompts from the Google C4 news dataset[1]. We include language models spanning a wide range of scales: parameter sizes varying from 125 million to 8 billion, and vocabulary sizes ranging in 32-262 thousands; for watermarking schemes, we consider Gumbel-max trick (Aaronson, 2023), Inverse transform (Kuditipudi et al., 2024), Red-green watermark (Kirchenbauer et al., 2023) and Permute-and-Flip watermark (Zhao et al., 2025). In each scenario, the first 50 tokens of a news article have been provided as inputs to the language models, and $n = 500$ output tokens are recorded. Among these 500 output tokens, there are two watermarked segments: 100-200 and

---

[1] https://www.tensorflow.org/datasets/catalog/c4

325-400. The specific tuning parameter choices for `WISER` are provided in §C. Table 1 showcases the results for the Gumbel watermarking scheme. It is evident that `WISER` outperforms all the other algorithms across all the metrics for each model. The detailed discussion, including the specific metrics used and additional results and insights, are provided in Appendix §C.1.

| Model Name | Vocab Size | Method | IOU | Precision | Recall | F1 | RI | MRI |
|---|---|---|---|---|---|---|---|---|
| facebook/opt-125m | 50272 | WISER | **0.944** | **1.000** | **0.995** | **0.997** | **0.984** | **0.979** |
| | | Aligator | 0.734 | 0.382 | 0.988 | 0.551 | 0.939 | 0.931 |
| | | Waterseeker | 0.672 | 1.000 | 0.802 | 0.890 | 0.864 | 0.850 |
| | | SeedBS-NOT | 0.479 | 0.730 | 0.625 | 0.673 | 0.844 | 0.823 |
| google/gemma-3-270m | 262144 | WISER | **0.896** | **0.965** | **0.960** | **0.962** | **0.953** | **0.950** |
| | | Aligator | 0.506 | 0.234 | 0.912 | 0.373 | 0.881 | 0.861 |
| | | Waterseeker | 0.645 | 0.968 | 0.775 | 0.861 | 0.851 | 0.836 |
| | | SeedBS-NOT | 0.362 | 0.610 | 0.478 | 0.536 | 0.753 | 0.704 |
| facebook/opt-1.3b | 50272 | WISER | **0.934** | **1.000** | **0.995** | **0.997** | **0.981** | **0.974** |
| | | Aligator | 0.497 | 0.235 | 0.920 | 0.375 | 0.892 | 0.871 |
| | | Waterseeker | 0.657 | 1.000 | 0.808 | 0.893 | 0.860 | 0.846 |
| | | SeedBS-NOT | 0.360 | 0.618 | 0.465 | 0.531 | 0.766 | 0.731 |
| princeton-nlp/Sheared-LLaMA-1.3B | 32000 | WISER | **0.939** | **1.000** | **0.998** | **0.999** | **0.983** | **0.978** |
| | | Aligator | 0.459 | 0.236 | 0.912 | 0.376 | 0.886 | 0.862 |
| | | Waterseeker | 0.659 | 1.000 | 0.812 | 0.897 | 0.862 | 0.847 |
| | | SeedBS-NOT | 0.278 | 0.520 | 0.388 | 0.444 | 0.731 | 0.699 |
| mistralai/Mistral-7B-v0.1 | 32000 | WISER | **0.909** | **1.000** | **0.998** | **0.999** | **0.975** | **0.961** |
| | | Aligator | 0.292 | 0.215 | 0.745 | 0.334 | 0.811 | 0.774 |
| | | Waterseeker | 0.621 | 1.000 | 0.765 | 0.867 | 0.840 | 0.824 |
| | | SeedBS-NOT | 0.240 | 0.442 | 0.320 | 0.371 | 0.657 | 0.593 |
| meta-llama/Meta-Llama-3-8B | 128256 | WISER | **0.926** | **1.000** | **0.988** | **0.994** | **0.977** | **0.975** |
| | | Aligator | 0.546 | 0.367 | 0.925 | 0.525 | 0.911 | 0.891 |
| | | Waterseeker | 0.570 | 1.000 | 0.720 | 0.837 | 0.814 | 0.791 |
| | | SeedBS-NOT | 0.379 | 0.620 | 0.515 | 0.563 | 0.778 | 0.741 |

Table 1: Results for Gumbel Watermarking

## 4.2 TIME COMPARISON

As established in §3.2, the proposed `WISER` algorithm achieves a computational complexity of $\approx O(n)$. Figure 3 provides empirical evidence supporting this theoretical claim and, in addition, compares the runtime behavior of `WISER` with other state-of-the-art methods. For this experiment, we randomly create an $n/6$-length watermarked segment using the Gumbel-max trick with NTP generated by Google's Gemma-3 model; block size was taken as $\lceil\sqrt{n}\rceil$ and $\rho = 0.1$. The results clearly indicate that `WISER` consistently outperforms competing approaches in terms of computational efficiency, emerging as the fastest among all methods considered in this study.

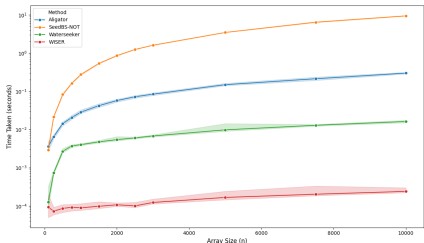

Figure 3: Time complexity (seconds) for various algorithms as a function of completion lengths ($n$). Y-axis is in log-scale, with 95% confidence interval shown in shades.

## 5 CONCLUDING REMARKS

In this paper, we introduced `WISER`, a first-of-its-kind algorithm for efficient and theoretically valid segmentation of watermarked intervals in mixed-source texts. By framing watermark localization as an epidemic change-point problem, we bridged a novel connection between classical statistical theory and a modern challenge in generative AI, and also designed a linear time algorithm with provable consistency guarantees, which were further confirmed by our extensive numerical experiments. Beyond the findings of this paper, it is also crucial to theoretically investigate the robustness of the proposed algorithm under human edits (Li et al., 2024a); as a roadmap, we have already included some relevant discussion in Appendix §A. Its applicability to multimodal (e.g. audio, image, video) settings (Qiu et al., 2024) also presents opportunities for future research.

## ETHICS STATEMENT

The research follows all ethical guidelines. No human data or ethically sensitive content is involved. All potential limitations and justifications are adequately addressed. We do not anticipate any negative impacts, and as such the paper does not include a dedicated speculative discussion of broader societal impacts.

## REPRODUCIBILITY STATEMENT

The datasets and the large language models were acquired from the open-source Huggingface library. All the relevant reproducible codes and figures, as well as the generated datasets can be found in the anonymous Github repository. All the theoretical results and assumptions are rigorously proved and validated in §D.

## AUTHOR CONTRIBUTIONS

All the authors contributed equally to this research.

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
