# OpenReview forum: "\texttt{WISER}: Segmenting watermarked region- an epidemic change-point perspective."
_ICLR.cc/2026/Conference — ICLR 2026 Conference Withdrawn Submission_

### Official Review · Reviewer_7SYE · 2025-10-23

**Soundness:** 2
**Presentation:** 2
**Contribution:** 2
**Rating:** 2
**Confidence:** 4

**Summary:**

This paper introduces WISER, a new algorithm for detecting watermarked segments in human-AI mixed text. The method models the task as an epidemic change-point detection problem and employs a coarse-to-fine strategy to achieve what the authors report as state-of-the-art accuracy and efficiency.

**Strengths:**

1. **Problem Significance**: The problem of identifying AI-generated content is critical and highly relevant.

2. **Impressive Results**: The reported performance metrics (e.g., IoU, F1) significantly outperform prior state-of-the-art methods.

**Weaknesses:**

1. **Questionable Novelty over WaterSeeker**: My primary concern is that the proposed method's pipeline appears identical to that of WaterSeeker (Pan et al., 2024). **The entire process (signal conversion → coarse blocking → merging/filtering → boundary enlargement → fine-grained localization) is the same**. The paper fails to adequately articulate the fundamental algorithmic differences that distinguish WISER, thus casting doubt on its novelty.

2. **Unexplained Performance Gains**:  Given the striking similarity to WaterSeeker, the paper does not provide a convincing explanation for WISER's substantially superior performance. Are the gains from a genuinely better model, or from hyperparameter tuning or a different experimental setup? Without a clear ablation study or analysis pinpointing the source of the improvement, the results are difficult to interpret.

3. **Minor Issue**: The Appendix should be merged with the main paper into a single PDF for reader convenience.

**Questions:**

1. Could you please provide a detailed, step-by-step comparison between WISER and WaterSeeker? Now I don't see much difference.

2. What specific design choice or component in WISER is responsible for the significant performance leap over WaterSeeker? Please justify this with targeted analysis or ablation studies.

---

### Official Review · Reviewer_FKr3 · 2025-10-30

**Soundness:** 2
**Presentation:** 2
**Contribution:** 2
**Rating:** 4
**Confidence:** 3

**Summary:**

This paper introduces **WISER**, a novel algorithm for localizing watermarked segments within a text generated by a mixture of human and watermarked Large Language Model output. The authors provide **theoretical guarantees** for WISER's consistency and its computational efficiency, claiming an O(n) runtime. Extensive **numerical experiments** on various LLMs and watermarking schemes demonstrate that WISER outperforms existing state-of-the-art methods in both accuracy and speed.

**Strengths:**

*   **Novel Perspective:** The epidemic change-point perspective is a creative and theoretically grounded approach to a modern problem in AI safety and content authentication.
*   **Theoretical Rigor:** The paper provides strong theoretical foundations, including finite-sample error bounds and proof of consistency, which are rare in the existing watermark segmentation literature.
*   **Computational Efficiency:** The design of the algorithm prioritizes speed, achieving linear time complexity (O(n)), making it practical for analyzing long texts.

**Weaknesses:**

*   **Assumption Strength:** The theoretical guarantees depend on assumptions like "Elevated Alternatives" (Assumption 2.2) and "Minimum Separation" (Assumption 3.1). The practical validity of these assumptions across all real-world scenarios (e.g., heavily paraphrased watermarks) could be a limitation.
*   **Clarity of Practical Deployment:** The process for estimating the baseline `µ0` (the mean of pivot statistics for human text) is not detailed. In practice, this is a non-trivial requirement and its accuracy is crucial for WISER's performance.

**Questions:**

1.  **Estimation of µ0:** How do you recommend estimating `µ0` (the expected value of the pivot statistic under the human-written distribution) in practice? Is a held-out dataset of human text required, and how sensitive is WISER to errors in this estimation?
2.  **Robustness to Paraphrasing:** The paper mentions robustness as a motivation. Could you present quantitative results on WISER's performance when the watermarked text has been paraphrased or lightly edited, comparing it to the baselines? This would directly address a key practical challenge.
3.  **Multilingual Generalization:** The evaluation appears to use English-language benchmarks. Have you tested WISER on text generated from multilingual LLMs or with non-English watermarks? Does the algorithm make any language-specific assumptions?

---

### Official Review · Reviewer_YpJn · 2025-10-31

**Soundness:** 2
**Presentation:** 3
**Contribution:** 2
**Rating:** 2
**Confidence:** 3

**Summary:**

This paper aims to identify and locate watermarked segments in mixed-source text by modelling watermark detection as an epidemic change-point problem. The authors theoretically validate their algorithm by deriving finite-sample error bounds and compare their method with existing approaches on self-constructed datasets. The presented results show that their method outperforms others.

**Strengths:**

1.	The paper provides theoretical analysis for the proposed algorithm under certain assumptions.
2.	The proposed algorithm is computationally efficient.
3.	Experimental results show the proposed method outperforms competing methods.

**Weaknesses:**

1.	Some assumptions in the theoretical part are too strong. For examples, Assumption 3.1 requires the distance between any two watermarked segments exceed a lower bound that grows with n^{1/2}log(n). This assumption becomes unrealistic when n is large (i.e., when the text is long), as segments can be much closer in practice. Moreover, the assumptions on manually chosen parameters, such as block size and threshold \rho, are also idealised.
2.	The method appears to depend heavily on the choice of b and \rho. The number of detected segments K is implicitly determined by the significance threshold \rho, which is manually fixed. Since the authors already know the true K for the test texts, tuning b and \rho may naturally lead to good performance.
3.	The experimental datasets are synthesized under the assumption (e.g. the spacing between watermarked segments are large) favourable to the proposed method. In addition, all reported results correspond to the case K=2, with segment locations fixed at 100–200 and 325–400. The method’s scalability to larger K or irregular segment lengths and spacing remains untested, which makes the experimental results less convincing.
4.	In each experiment, the watermarked segments are generated by the same LLM. The authors did not consider the scenario where watermarked segments are produced by different LLMs with different watermarking parameters or schemes.
5.	The robustness of the proposed method against common attacks (e.g., cross-model paraphrasing or sentence deletions) has not been evaluated.

**Questions:**

1.	The authors recommend b \in (\sqrt(n), 3\sqrt(n)). However, this range can be very wide when n is large. Do the authors have any strategies for the choice of b and \pho if both n and K are very large? If K is unknown and varies, can the algorithm handle the text without manual tuning?
2.	Have the authors tested cases with larger K, random segment lengths and small spacing?
3.	Does the method work on single text containing segments watermarked by different LLMs or watermarking schemes?
4.	With the same hyperparameters, is the method robust to cross-model paraphrasing and sentence deletions?

---

### Official Review · Reviewer_scn3 · 2025-11-04

**Soundness:** 2
**Presentation:** 2
**Contribution:** 2
**Rating:** 6
**Confidence:** 3

**Summary:**

This paper proposes a new algorithm called WISER to locate multiple watermarked segments in text from mixed sources. It redefines the watermark localization problem as an epidemic change-point detection problem, combining classical statistical theory with the challenges of modern generative AI. WISER achieves significant improvements in computational efficiency (with linear time complexity O(n)) and theoretical consistency, making it the first watermark segmentation algorithm with complete theoretical guarantees under general conditions.

**Strengths:**

1. Innovative Perspective: It combines the watermark segmentation problem with epidemic change-point theory, providing a novel theoretical framework.
2. Theoretical Contribution: It offers rigorous theoretical guarantees, proving the algorithm’s consistency in estimating the number and positions of watermarked segments.
3. Computational Efficiency: With a linear time complexity of O(n), the algorithm outperforms existing methods (such as Aligator and Waterseeker) significantly.
4. Experimental Validation: Extensive experiments verify the algorithm’s superiority across different watermarking schemes and language models, using multiple metrics including IOU, Precision, and Recall.
5. Strong Practicality: It addresses the issues of slow processing speed for long texts and lack of theoretical support in existing methods, presenting a feasible solution.

**Weaknesses:**

1. The algorithm’s performance relies on parameter selection. Although the paper mentions a robustness interval, further adjustments may be required in practical applications.
2. Some theoretical assumptions (e.g., Assumption 2.2 and 3.1) may be difficult to fully satisfy in real-world scenarios, especially when texts are frequently edited.
3. While the paper mentions applicability to human-edited texts (Appendix A), it does not fully discuss the algorithm’s limitations in complex real-world scenarios in the main body.

**Questions:**

1.How to adaptively select key parameters when there is a lack of prior information? Is there an automatic parameter tuning mechanism?
2. The experiments are mainly based on English text. Is the algorithm applicable to other languages or scenarios with different token distributions?
3. Although the algorithm has a complexity of O(n), is the actual running time still acceptable for extremely long texts (e.g., n > 10^6)?
4. How robust is the algorithm against targeted watermark tampering or adversarial attacks?

---

### Note · Authors · 2025-11-12

I have read and agree with the venue's withdrawal policy on behalf of myself and my co-authors.